# Identification and Analysis of the Mechanism of Stem Mechanical Strength Enhancement for Maize Inbred Lines QY1

**DOI:** 10.3390/ijms25158195

**Published:** 2024-07-27

**Authors:** Yumeng Yang, Jianing Mu, Xiaoning Hao, Kangkang Yang, Ziyu Cao, Jiping Feng, Runhao Li, Ning Zhang, Gongke Zhou, Yingzhen Kong, Dian Wang

**Affiliations:** 1College of Agronomy, Qingdao Agricultural University, Qingdao 266109, China; 20222101061@stu.qau.edu.cn (Y.Y.); 20222201011@stu.qau.edu.cn (J.M.); haoxn0530@126.com (X.H.); yangkk-ceres@ttc-cert.com (K.Y.); 20232101004@stu.qau.edu.cn (Z.C.); 20232201005@stu.qau.edu.cn (J.F.); 20232101024@stu.qau.edu.cn (R.L.); 20232201034@stu.qau.edu.cn (N.Z.); 2College of Landscape Architecture and Forestry, Qingdao Agricultural University, No. 700 Changcheng Road, Chengyang District, Qingdao 266109, China; zhougk@qau.edu.cn; 3Academy of Dongying Efficient Agricultural Technology and Industry on Saline and Alkaline Land in Collaboration with Qingdao Agricultural University, Dongying 257000, China

**Keywords:** maize, stem mechanical strength, cell wall, biofuel

## Abstract

Enhancing stalk strength is a crucial strategy to reduce lodging. We identified a maize inbred line, QY1, with superior stalk mechanical strength. Comprehensive analyses of the microstructure, cell wall composition, and transcriptome of QY1 were performed to elucidate the underlying factors contributing to its increased strength. Notably, both the vascular bundle area and the thickness of the sclerenchyma cell walls in QY1 were significantly increased. Furthermore, analyses of cell wall components revealed a significant increase in cellulose content and a notable reduction in lignin content. RNA sequencing (RNA-seq) revealed changes in the expression of numerous genes involved in cell wall synthesis and modification, especially those encoding pectin methylesterase (PME). Variations in PME activity and the degree of methylesterification were noted. Additionally, glycolytic efficiency in QY1 was significantly enhanced. These findings indicate that QY1 could be a valuable resource for the development of maize varieties with enhanced stalk mechanical strength and for biofuel production.

## 1. Introduction

Maize, a critical global food crop, is extensively cultivated. Increasingly, extreme weather events affect maize production, with lodging and stem breakage presenting major challenges [1]. The stem’s chemical composition, largely determined by its cell wall components, is crucial for lodging resistance [2]. Cellulose, a primary cell wall constituent, creates a network that bolsters mechanical strength and stability. This supports the plant, maintaining its structure and posture. Structural integrity is essential for preserving cell shape and withstanding external forces [3]. Microscopically, the cellulose content and quality significantly influence the stem’s mechanical strength and its resistance to lodging [4]. Thus, improving the cellulose content and quality in maize stems is essential for enhancing lodging resistance and has significant implications for the recycling and reuse of agricultural waste.

The plant cell wall, unique to plant cells, performs multiple functions in plant growth and development. It provides structural support, maintains the plant’s external morphology, aids in water and nutrient transport, and enables plants to adapt to their environment and resist pest invasions [5]. It also plays a significant role in plant growth, development, and signal transduction [6]. The cell wall is categorized into two types: primary and secondary. The primary cell wall is essential for determining plant morphology; it is relatively thin and elastic, permitting cell growth and extension. Composed mainly of cellulose microfibrils, pectin, hemicellulose, and proteins, it is engineered for strength from cellulose microfibrils, while pectin and hemicellulose provide flexibility and plasticity [7]. The secondary cell wall, which develops externally to the primary cell wall after cell growth ceases, provides robust mechanical support and facilitates essential water transport for plant growth. It also serves as a barrier against various biotic and abiotic stresses [8]. In addition to cellulose, hemicellulose, and pectin, the secondary cell wall contains a higher proportion of lignin, a complex organic polymer that enhances cell wall strength. The ratio of lignin to cellulose significantly influences the physical properties of the plant stem, affecting growth performance and agricultural productivity. Managing this ratio is vital for enhancing crop lodging resistance and can influence crop processing quality and the efficiency of end-product utilization [9].

Maize, widely cultivated and a significant producer of straw biomass, is a vital bioenergy source. However, in many countries, straw is not effectively processed. Common methods include returning straw to the field or burning it. Returning straw is inefficient and expensive, and excessive amounts can degrade soil aeration and water infiltration, stunting crop growth or causing soil acidification and increased pest risks. Burning straw, a rudimentary method, severely damages the environment [10]. Currently, biological treatment technology is the only method achieving resource recycling in straw processing. By transforming agricultural waste like straw into ethanol fuel through bioethanol production technology, we reduce reliance on traditional energy sources and curb greenhouse gas emissions. However, bioethanol production encounters significant obstacles: the raw materials are sourced from fields, leading to high collection and transportation costs and a constrained supply. Moreover, converting cellulose in straw into fermentable sugars is a complex and expensive process, driving up production costs [11]. The lignocellulosic component of maize straw is crucial for biofuel production. Lignin compromises cell wall digestibility and saccharification. Therefore, straw with high cellulose and low lignin content is preferable, boosting saccharification efficiency and lowering biofuel production costs [12,13,14,15,16].

This study focuses on maize inbred lines B73 and QY1, exploring their phenotypic traits, stem mechanical strength, anatomical structure, and cellular composition differences. Through transcriptome sequencing analysis, we identify key candidate genes and regulatory networks that influence cell composition differences and affect stem mechanical strength. This research aims to provide a theoretical foundation for breeding high-yield, high-lodging-resistant maize varieties with improved cellulose content that improved straw recycling.

## 2. Results

### 2.1. Enhanced Mechanical Strength in Maize Inbred Line QY1

We examined the morphological phenotypes of mature maize inbred lines B73 and QY1. Although plant height was consistent between B73 and QY1, the fifth internode stem of QY1 was significantly thicker by 18.50% (Figure 1B,C). This increase may influence mechanical strength and indicate compositional differences, steering our research focus. We then measured the stem mechanical strength of B73 and QY1, including the stem puncture strength, compressive strength, and leaf tensile strength. Our results showed that, compared to B73, both upper and basal stem puncture strengths of QY1 rose significantly by 25.00% and 145.18%, respectively. Additionally, leaf breaking forces and stem bending strength increased by 20.05% and 42.77% (Figure 1D–G). In conclusion, the mechanical strength of QY1 stems was significantly enhanced, resulting in a significant improvement in lodging resistance. This increase was likely due to changes in the structure and composition of the stem cell wall. Therefore, QY1 not only exhibited stem thickening but also enhanced mechanical strength.

### 2.2. Increased Cell Wall Thickness in QY1 Stem Vascular Bundles

To assess if the enhanced mechanical strength in QY1 stems relates to changes in cell wall structure, we performed paraffin embedding and sectioning on the basal and upper stems of B73 and QY1, examining their cross-sectional structures microscopically.

The analysis revealed that, in comparison to B73, QY1 had a greater number of epidermal cell layers in both basal (Figure 2A–D) and upper stems (Figure 2F–I), with vascular bundles densely packed around the stem periphery. Both epidermal and vascular bundle cell walls in QY1 were significantly thicker, and the vascular bundle area increased. Statistical analysis showed a significant increase in cell wall thickness of 53.47% in epidermal cells and 46.87% in vascular bundle cells of the basal stem (Figure 2E), and of 15.10% and 8.44% in those of the upper stem, respectively (Figure 2J). From basal to upper stem, the cell walls of QY1 were notably thickened, particularly in the basal sections. These findings suggest that cell wall thickening and increased vascular bundle area contribute to the improved mechanical strength and lodging resistance of QY1 stems.

### 2.3. Increased Cellulose Content and Decreased Lignin Content in QY1 Stalks

Upon examining the microstructure of the stem, notable changes in cell wall thickness were observed in QY1, indicating potential compositional changes. We extracted alcohol-insoluble residue (AIR) from both B73 and QY1 stems, analyzing the contents of cellulose, lignin, and monosaccharides.

Cellulose content in QY1 showed a significant increase of 16.09% compared to B73 (Figure 3A). High-performance liquid chromatography (HPLC) analysis of monosaccharide composition (Figure 3B) revealed increases in glucose (Glc) of 102.50% and galactose (Gal) of 57.90%. Conversely, there were decreases in xylose (Xyl) of 17.56%, glucuronic acid (GlcA) of 31.24%, and arabinose (Ara) of 17.83% within QY1. We also assessed the crystallization of cellulose, which showed a reduction in QY1 compared to B73 (Figure 3C).

Fresh stem sections stained with phloroglucinol-HCl of both B73 and QY1 highlighted the epidermal and vascular bundle regions. QY1 displayed lighter staining, indicative of a lower degree of lignification (Figure 3D–G). A significant reduction in lignin content was confirmed in QY1 (Figure 3H). Based on these findings, we conclude that the enhanced mechanical strength of QY1 stems is due to the increased cellulose and glucose content. These data suggest a correlation between the enhanced mechanical strength of QY1 stems and the increase in cellulose content.

### 2.4. Increased Saccharification Efficiency in Inbred Line QY1

Given lignin’s critical impact on the utilization of cellulose biomass in maize stems, we measured the cell wall digestibility of B73 and QY1. The results revealed a significant improvement in the enzymatic hydrolysis efficiency of QY1 compared to B73. Under both untreated and dilute acid-treated conditions, QY1 experienced increases of 5.4% and 8.4%, respectively (Figure 4A). Additionally, saccharification efficiency improved from 16.6% and 30.6% in wild-type maize to 21.2% and 37.4% in QY1 (Figure 4B). The increased sugar output in QY1 can be attributed to the significant rise in cellulose content. Moreover, the enhanced cell wall saccharification efficiency is likely due to the reduced influence of lignin on the digestibility enzymes. In conclusion, with sufficient digestive enzymes, QY1 demonstrated remarkable improvements in cell wall digestibility and total soluble sugar production.

### 2.5. QY1 Influences Stem Strength through Gene Regulation

To identify key genes regulating stem development in QY1, we selected three mature plants of each variety, B73 and QY1, showing healthy growth for transcriptome sequencing of the fifth internode at the base. Transcriptome analysis revealed 8580 differentially expressed genes (DEGs) between QY1 and B73, with 3586 upregulated and 4994 downregulated (Appendix A). Significant differential gene expression was noted between the two varieties in the heatmap results.

In the biological process category, DEGs in QY1 and B73 were significantly enriched in DNA replication, oxidative stress response, auxin response, and chemical response. In the cellular component category, DEGs were concentrated in the extracellular region, cell periphery, and cell wall. In the molecular function category, DEGs were predominantly involved in heme binding, tetrapyrrole binding, polysaccharide binding, carbohydrate binding, and transferase activity (Figure 5A, Appendix A). GO enrichment analysis indicated that these DEGs are primarily involved in cell wall processes and may regulate QY1’s lodging resistance at the molecular level.

GO enrichment analysis also highlighted 18 DEGs related to cell wall composition, analyzed via a heatmap (Figure 5B), showing 11 upregulated and 7 downregulated genes. According to annotations on the Maize GDB website (Table 1), these genes are associated with pectin methylation and xylan synthesis. Expression patterns from the Maize eFP website revealed that genes *Zm00001eb187870* and *Zm00001eb226490* exhibited higher expression in the stem (Appendix A) and were upregulated by tenfold and sixfold in QY1, respectively, associated with xylan transferase. Xyloglucan (XyG), a β-(1,4)-glucan commonly binding to cellulose, suggests these genes are key regulators of QY1 stem mechanical strength. RNA-seq results were validated by qPCR for eight cell wall-related genes, corroborating the transcriptome data (Figure 5C). Additionally, eight CesA-related DEGs were identified, with CesA being a critical enzyme in cellulose synthesis. The enrichment of CesA family genes in plants can increase cellulose content, with *ZmCesA2* and *ZmCesA3* being upregulated by 19-fold and 235-fold, respectively (Table 2). This suggests that the upregulation of CesA genes in QY1 might influence cellulose content, thereby affecting lodging resistance. *ZmCesA2* and *ZmCesA3* are likely key genes in regulating stem development.

KEGG analysis revealed significant enrichment of DEGs in the phenylpropanoid biosynthesis pathway, with pathway and heatmap analyses showing predominantly downregulated DEGs related to phenylalanine biosynthesis in QY1 (Appendix A). DEGs associated with enzymes PAL, 4CL, and CCR in the initial reaction of the pathway were all downregulated. These DEGs affect lignin biosynthesis, and biochemical analyses detected a significant decrease in lignin content, suggesting that this reduction is regulated by these DEGs.

Previous studies have established that transcription factors regulate plant cell wall development. Key transcription factors, including ERF, NAC, MYB, and WRKY, are crucial in cell wall biosynthesis regulation. Among the DEGs, significant upregulation and downregulation of these transcription factors were noted (Appendix A). These transcription factor-associated DEGs may be integral to regulating the cell wall development of QY1 stems. Together, the expression changes in genes related to cell wall synthesis are consistent with alterations in cell wall composition and thickness.

### 2.6. Changes in Pectin Methylation and Methylesterase Activity in QY1

Galacturonic acid (HG), a pectin component synthesized in the Golgi apparatus, is initially highly methyl-esterified. After synthesis, HG is secreted into the cell wall where it is demethylated by PME to become functional.

Previous research has shown that Arabidopsis *PME35* influences stem development. In both QY1 and B73 lines, numerous cell wall DEGs are related to PME. Evolutionary analysis showed high homology between *Zm00001eb11250* (*ZmPME12*) and Arabidopsis *AtPME35* (Figure 6A), with *ZmPME12* upregulated in QY1. We measured PME activity and the degree of pectin methylation (DME) in both lines (Figure 6B,C). The results demonstrated that QY1 had a significant reduction in PME activity, leading to decreased demethylesterification capacity and a substantial 16.66% increase in DME. These findings suggest that increased pectin methylation may correlate with enhanced cell wall mechanical strength.

## 3. Discussion

This study identified a maize inbred line, QY1, with notably rigid stems through selfing populations. We analyzed its growth parameters, morphological structures, cell wall composition, and transcriptome to explore variations in components and phenotypes that contribute to stem rigidity. The aim was to identify key genes for stem development, providing a foundation for breeding high-quality, lodging-resistant varieties and improving straw utilization efficiency.

### 3.1. Increased Cell Wall Thickness in QY1 Enhances Mechanical Strength of Stalks

Previous studies have indicated that plant height and stem thickness are critical for lodging resistance, with shorter and thicker stems offering greater protection. For instance, in rice, the *myb110* mutant showed increased lodging resistance due to thicker stems [17]. Mechanical strength is essential for the structural stability of cereal crops and correlates with lodging resistance. Earlier findings have noted that brittle culm mutants *bc3* in rice and *bk4* in maize exhibited greater brittleness, significantly diminished mechanical strength, and reduced lodging resistance [18,19].

We assessed the growth parameters of B73 and QY1 and observed that QY1 featured significantly thicker stems. These robust stems were more resistant to bending under stress, enhancing lodging resistance. Puncture strength tests showed that QY1’s basal and upper stems required greater force to pierce compared to B73. During compressive strength testing, while B73 stems fractured under certain pressure, QY1 stems demonstrated a smaller bending angle and required greater pressure to lodge.

Structural changes also impact mechanical strength. In the brittle-stem maize mutant *bk2*, a reduction in peripheral vascular bundles and thinner cell walls in sclerenchyma cells beneath the epidermis and around the vascular bundles resulted in decreased mechanical strength [20]. In rice, the *erf34* mutant exhibited weak stem mechanical strength and broke easily, whereas overexpression of *ERF34* led to thickened secondary walls, improved mechanical strength, and increased lodging resistance [21].

Microscopic examinations of stem cross-sections from B73 and QY1 revealed that, compared to B73, QY1 had more epidermal cell layers, tightly arranged vascular bundles at the tissue periphery, and larger vascular bundle areas. The increased size of these vascular bundles in QY1 supports the enhanced transport of water, nutrients, and photosynthetic products, strengthening plant support. Statistical analysis indicated that the cell walls of epidermal and vascular bundle cells in the basal and upper stems of QY1 were noticeably thicker. This enhancement bolstered the support strength of QY1 stems, reinforcing plant resilience under adverse conditions.

In conclusion, the increased vascular bundle area and thickened cell walls in QY1 contribute to enhanced stem mechanical strength and improve lodging resistance under stressful conditions. Consequently, QY1’s superior stem mechanical strength and excellent lodging resistance render it a valuable candidate for breeding lodging-resistant maize.

### 3.2. Significant Changes in QY1 Cell Wall Components

Polysaccharides, crucial components of plant cell walls, play vital roles in processes such as growth, development, signal transduction, and defense responses. In secondary cell walls, cellulose predominantly determines cell wall mechanical properties [22]; a deficiency in cellulose can lead to xylem collapse and reduced plant strength [23]. Cellulose synthase is essential for cellulose biosynthesis. Previous studies have shown that reduced mechanical strength in rice brittle culm mutants *bc3* and *bc15* resulted from lower cellulose content and altered wall structures [18,24]. The rice *myb110* mutant displayed an increased stem diameter and bending resistance, alongside a reduction in stem lignin content but enhanced lodging resistance [17]. Compared to the wild type, the rice mutant *fc19* had decreased xylose, hemicellulose, and cellulose but increased arabinose and lignin content, improving lodging resistance [25]. Transcriptome results from this experiment showed cellulose synthase-related genes were upregulated, primarily *CesA2* and *CesA3*, with *CesA3* upregulated by 236-fold. In Arabidopsis, *AtCesA2* and *AtCesA3* were linked to primary cell wall cellulose synthesis, and in *Atcesa2* mutants, there was a decrease in seed coat cellulose. *AtCesA4*, *AtCesA7*, and *AtCesA8* were involved in secondary cell wall synthesis, with *CesA3* playing a crucial role in forming functional CesA complexes [26]. We hypothesize that *ZmCesA2* and *ZmCesA3* may be key genes in regulating cellulose synthesis in QY1 stems and overall stem development.

Phenylpropanoid metabolism is a key secondary metabolic pathway in plants. Analysis of phenylpropanoid-related DEGs demonstrated that the genes encoding the enzymes PAL, C4H, and 4CL were all downregulated in QY1. Biochemical pathway assessments revealed a significant reduction in lignin content in QY1, likely due to the regulation of these genes.

Furthermore, the degree of pectin methylation influences the rheological properties of the cell wall, which are vital for plant growth, development, and stress responses. Previous studies in Arabidopsis showed that *PME35* regulated stem mechanical strength; the *pme35* mutant exhibited reduced PME activity, decreased demethylation capacity, and increased pectin methylation, leading to more flexible stems [27]. Among the cell wall-related DEGs, *ZmPME12*, sharing high homology with *AtPME35*, was significantly upregulated threefold in QY1. PME activity and DM measurements in B73 and QY1 indicated that PME activity was significantly reduced in QY1, while pectin methylation was significantly increased. We hypothesize that the mechanical strength of QY1 stems is influenced by the coordinated action of these PME-related genes, with *ZmPME12* likely playing a key role in regulating stem strength.

According to our transcriptome results, we identified several genes previously studied in other plants. In maize, *MYB55* contributes to secondary cell wall formation. In C4 plants such as maize and sorghum, *GLK2* is predominantly expressed in the vascular bundle sheath, and *ZmGLK2* in maize enhances photosynthetic performance and yield. In Arabidopsis, the GLK1/2-WRKY40 transcriptional module negatively regulated ABA responses [28]. Transcriptome sequencing revealed significant upregulation of transcription factors *MYB55* and *GLK2* in QY1, suggesting these genes play crucial roles in QY1’s stress resistance. Additionally, studies have suggested that *NAC100* in wheat activates transcriptional pathways for both starch and gluten synthesis, with multiple transcription factors co-regulating these pathways [29]. The maize gene *NAC100* was upregulated by 34-fold in QY1, highlighting its potential role in regulating stem development. Further research is needed to explore how these genes affect maize stem development.

### 3.3. Application of QY1 to Agricultural Production

By analyzing the stem cell wall components of B73 and QY1 maize, we observed a notable twofold increase in glucose content associated with cellulose synthesis in QY1 stems. This increase in cellulose content likely enhances the hardness of QY1 stems. Additionally, a decrease in crystallized cellulose, a significant reduction in lignin content, and a decline in lignification were noted. While the glucose content in the monosaccharide components markedly increased, the xylose content significantly decreased.

These alterations not only strengthen the mechanical properties of the stems but also broaden their use in agricultural production. The boost in cellulose content may make QY1 maize stems more resistant to decay, thereby bolstering their resilience against wind, pests, and lodging. Furthermore, when the straw is recycled into the soil as organic fertilizer, its high cellulose content could enhance the growth and activity of soil microorganisms, enhancing soil structure and boosting organic matter content. This leads to improved maize yield and quality, reduced losses due to climate, pests, and diseases, and augmented soil fertility and moisture retention, benefitting agricultural production.

The most immediate impact is the marked improvement in saccharification efficiency attributable to the changes in cellulose content, leading to increased bioethanol production efficiency and economic gains in maize production. As cellulose is the primary raw material for bioethanol production, its degradation releases glucose that can be fermented into ethanol. The increase in cellulose, along with adequate digestive enzymes, enables more efficient degradation into sugars. Additionally, previous studies have shown that lignin provides cross-linking protection to cellulose and hemicellulose, limiting their hydrolytic conversion efficiency. Moreover, previous studies have shown that lignin limits the conversion efficiency of lignocellulosic biomass to glucose, thereby reducing bioethanol production efficiency and increasing production costs. Consequently, lignin has become a significant barrier to producing bioethanol from corn stalks, and reducing the lignin content in corn stalks represents a critical breakthrough for achieving efficient bioethanol production [30]. As the lignin content at the basal stem is highest, the saccharification efficiency of basal stems is lower than other tissues. However, the saccharification efficiency of QY1 is greatly enhanced. This provides higher-quality raw materials for the production of second-generation bioethanol and markedly improves the utilization rate of maize straw [12]. In summary, the differences in stem cell wall components, particularly the increase in cellulose content, could provide valuable insights and directions for technological advancements in maize straw bioethanol production, supporting the development of the bioenergy industry, reducing reliance on traditional fossil fuels, and reducing crucial technical support for advancing sustainable agriculture.

## 4. Materials and Methods

### 4.1. Experimental Material

The maize inbred lines B73 and QY1 were used in this study. QY1 was derived from a large-scale population established through mixed pollination, with B73 as the maternal line and varied pollen sources as the paternal. From this population, we selected an inbred line, named QY1, which exhibited significantly enhanced mechanical strength in stalks.

B73 and QY1 were cultivated in the fields of Shandong and Hainan provinces in China and in the greenhouse at Qingdao Agricultural university. The greenhouse conditions were maintained at 25℃, with 30–40% ambient humidity, a light/dark cycle of 16/8 h, and soil composed of equal parts domestic soil, imported soil, and vermiculite.

### 4.2. Determination of Plant Morphology and Mechanics

Ten plants each of B73 and QY1, grown in the greenhouse for approximately 60 days and in good condition, were selected for mechanical testing. The fifth leaf from the base was used for the leaf tension test to determine the maximum tension before breakage. The basal stem was subject to a stress test to assess the maximum pressure it could withstand before breaking. Additionally, the fifth internode of the basal stem was used for a stalk puncture strength test to measure the maximum force of probe penetration.

Leaf tension and stem pressure resistance were measured using a CTM2500 electronic universal testing machine (Xieqiang Instrument, Shanghai, China). Stalk puncture strength was tested using a culm strength tester (Zhejiang topu yunnong Technology Co., Ltd., Hangzhou, China).

### 4.3. Microscopy Analysis

The fifth internode of 60-day-old B73 and QY1 plants was sectioned into 0.5 cm segments using a razor blade, fixed in 4% (*w*/*v*) paraformaldehyde, and embedded in paraplast as described by Tang et al. (2020). Stem sections were cut with a Leica RM 2235 microtome, stained with 0.1% (*w*/*v*) toluidine blue O (Sigma, St. Louis, MO, USA) for 5 min, and examined under a microscope (AXIO Scope A1; ZEISS, Oberkochen, Germany).

### 4.4. Phloroglucinol–HCl Stain

The fifth internode of 60-day-old maize from lines B73 and QY1 was sectioned freehand to produce thin slices of stem cross-sections (20–40 μm thick). The sections were stained for 5 min in Phloroglucinol (Sigma) in 20% HCl, rinsed with water, then covered with a coverslip and examined under a microscope (AXIO Scope A1; ZEISS) to assess the degree of lignification [31].

### 4.5. Extraction of Alcohol-Insoluble Residue

The fifth internode of maize inbred lines B73 and QY1, grown in the greenhouse for 60 days, was harvested and ground in a ball mill. The milled tissue was washed with pre-warmed 70% (*v*/*v*) ethanol at 70 °C, vortexed, and pelleted by centrifugation at 14,000× *g* for 10 min. The pellet was subsequently suspended twice in a chloroform mixture (1:1, *v*/*v*), shaken for 60 min at room temperature, and centrifuged at 14,000× *g* for 10 min. After repeated centrifugation under the same conditions, pellets were resuspended twice in 1 mL of 80% (*v*/*v*) acetone and spun at 14,000× *g* for 5 min. The supernatants were discarded, and the pellet containing the AIR was dried. To remove starch, the AIR was treated with pullulanase M3 (0.5 U mg^−1^, Megazyme, Bray, County Wicklow, Ireland) and α-amylase (1 U mg^−1^, Sigma, St. Louis, MO, USA) in 0.1 M NaOAc buffer for 12 h at 37 °C [14]. 

### 4.6. Determination of Cellulose and Lignin Content

Cellulose content was determined using the method described previously (Qi et al. 2015). In brief, AIR was hydrolyzed by trifluoroacetic acid (TFA) at 120 °C for 120 min. The TFA resistant materials were then treated with Updegraff reagent (acetic acid: nitric acid: water, 8:1:2, *v*/*v*) at 100 °C for 30 min, and the resulting pellets were completely hydrolyzed using 67% H_2_SO_4_ (*v*/*v*). The released glucose was measured using a glucose assay kit (Cayman Chemical, Ann Arbor, MI, USA) with a dehydration factor of 0.9. Total lignin content was determined by the AcBr method [32,33].

### 4.7. Determination of Pectin Methylesterase (PME) Activity

PME activity in maize inbred lines B73 and QY1 was quantified using the PECTOPLATE assay as previously described [34]. Total proteins were extracted from the fifth internode of 60-day-old maize in the presence of 1 M NaCl, 12.5 mM citric acid, 50 mM Na_2_HPO_4_, 0.02% (*w*/*v*) sodium azide, and 1:100 (*v*/*v*) protease inhibitor (535142; Millipore, Burlington, MA, USA), pH 6.5. The supernatant was collected, and protein concentration was determined using the Bradford protein assay [35]. Protein samples (2 µg) were incubated at 30 °C for 16 h and stained with 0.05% (*w*/*v*) Ruthenium Red (R2751; Sigma, St. Louis, MO, USA) for 30 min. The plates were then destained with several washes of water, and the areas of the fuchsia-stained haloes, indicating the demethylesterification of pectin, were measured using ImageJ (version number: 1.53k) software [36]. A standard curve was generated using commercially available PME (P5400; Sigma, St. Louis, MO, USA) to calculate PME activity in the protein extracts.

### 4.8. Determination of the Degree of Pectin Methylesterification (DME)

The pectin methylesterification (DME) assay was performed as described by Lionetti (2017). Briefly, 2 mg AIR was saponified by suspending it in 30 µL of water and 10 mL of 1 M NaOH and incubated for 1 h at room temperature. Afterward, 10 µL HCl was used to neutralize the mixture. The sample was centrifuged to obtain the supernatant, to which 50 µL of alcohol oxidase was added. The mixture was then incubated for 15 min at room temperature on a shaker. Thereafter, 100 mL of a mixture containing 0.02 M 2,4-pentanedione in 2 M ammonium acetate, and 0.05 M acetic acid was added to each well. The samples were incubated at 68 °C for 10 min, cooled, and then measured at 412 nm in a microplate reader (SPECTRAMAX PLUS384; Molecular Devices, San Jose, CA, USA). The DME is expressed as a MeOH-to-uronic acid molar ratio (%).

### 4.9. Matrix Polysaccharide Composition Analysis

The matrix polysaccharide composition of B73 and QY1 was analyzed using TFA-hydrolyzed materials as previously described [37]. The released monosaccharides were derivatized with 1-phenyl-3-methyl-5-pyrazolone (PMP), and the derivatives were analyzed by HPLC.

### 4.10. Determination of Cell Wall Saccharification Efficiency

AIR samples taken from 60-day-old B73 and QY1 maize basal stems were analyzed for saccharification efficiency. The AIR was pre-treated with 15% dilute H_2_SO_4_ at 121 °C for 60 min. After rinsing with Milli-Q water, the samples were incubated with a mixture of cellulase and cellobiase for 72 h. The phenol-sulfuric acid assay was used to measure soluble sugar content before and after enzymatic digestion, recorded as the total sugar content and released sugar content, respectively. Cell wall enzymatic hydrolysis efficiency was calculated using the formula: cell wall saccharification efficiency (%) = (released sugar content/total sugar content) × 100. Additionally, the total solubilized sugar yield from enzymatic hydrolysis was calculated as follows: solubilized sugar yields (g plant^−1^) = cell wall carbohydrate yield of the biomass (g plant^−1^) × saccharification efficiency [38].

### 4.11. Transcriptome Analysis

Three plants each of maize B73 and QY1, grown and matured in Hainan, were selected and labeled B73-1, B73-2, B73-3, QYI-1, QY1-2, and QY1-3. The fifth internode of the basal stem was flash-frozen in liquid nitrogen, stored in 50 mL RNase Free centrifuge tubes, and sent for transcriptome sequencing at Beijing Novozymes Bioinformatics Co. (Beijing, China)The data accession number is PRJNA1127697.

To verify the accuracy of the RNA-seq results, eight DEGs were chosen for real-time fluorescence quantitative PCR (qRT-PCR) verification. The reliability of the transcriptome data was assessed by comparing the results from both methods. qRT-PCR primers were designed using Primer blast at NCBI (Appendix A).

### 4.12. Data Analysis

Data processing and plotting were performed using Microsoft Excel 2019; statistical significance was analyzed using SPSS software(SPSS 25).

## Figures and Tables

**Figure 1 ijms-25-08195-f001:**
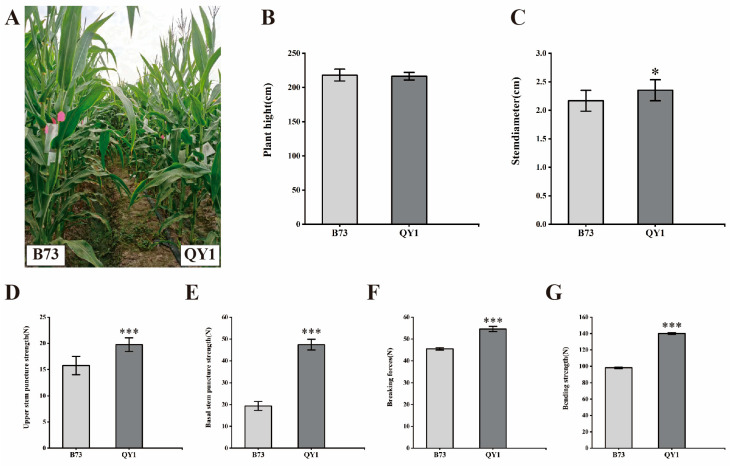
Phenotypes and physical properties of B73 and QY1 plants. (**A**) Phenotype of B73 and QY1. (**B**) Plant height and basal stem diameter (**C**) of B73 and QY1. Puncture strength of the stem (**D**,**E**), leaf breaking forces (**F**), and stem bending strength (**G**) of B73 and QY1. * (*p* < 0.05), *** (*p* < 0.001).

**Figure 2 ijms-25-08195-f002:**
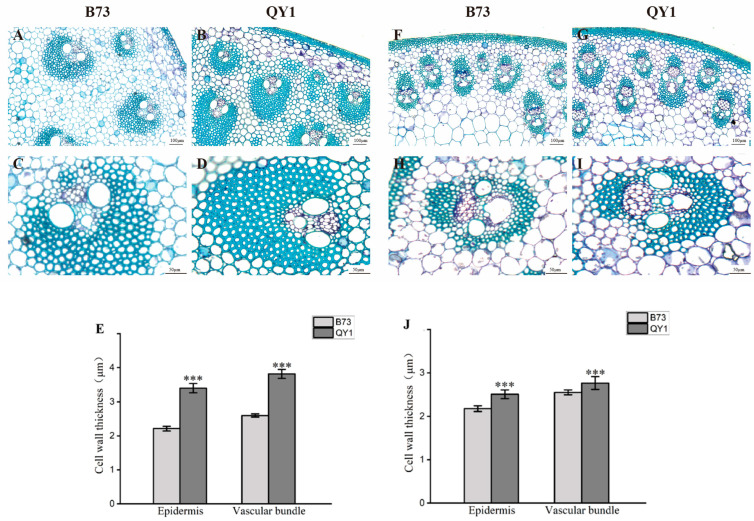
Microscopic structure analysis of B73 and QY1 stems. Cross-sections and vascular bundles of B73 and QY1 basal (**A**–**D**) and upper stems (**F**–**I**) stained with toluidine blue. Statistical analysis of cell wall thickness in outer epidermal and vascular bundle cells of basal and upper stems in B73 and QY1 (**E**,**J**), *** (*p* < 0.001).

**Figure 3 ijms-25-08195-f003:**
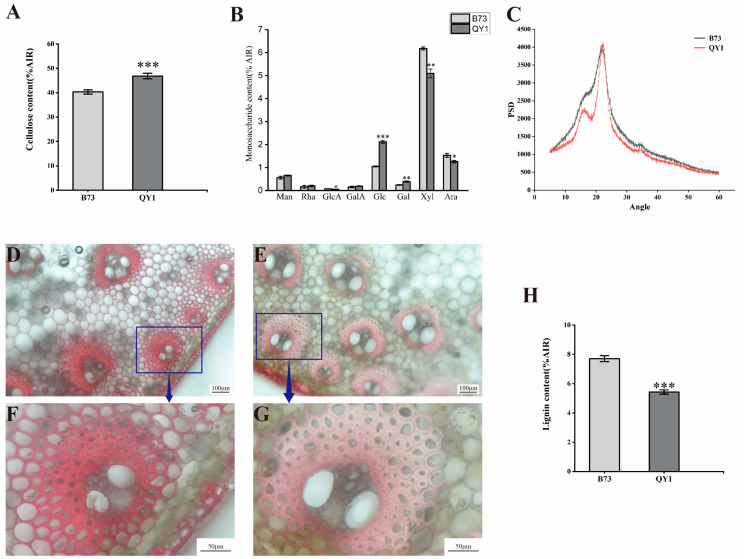
Analysis of cell wall composition in B73 and QY1 internodes. Cellulose content (**A**), matrix polysaccharide composition (**B**), crystalline cellulose (**C**), phloroglucinol-HCl staining (**D**–**G**), and lignin content (**H**) of stem cell wall residues. * (*p* < 0.05), ** (*p* < 0.01), *** (*p* < 0.001).

**Figure 4 ijms-25-08195-f004:**
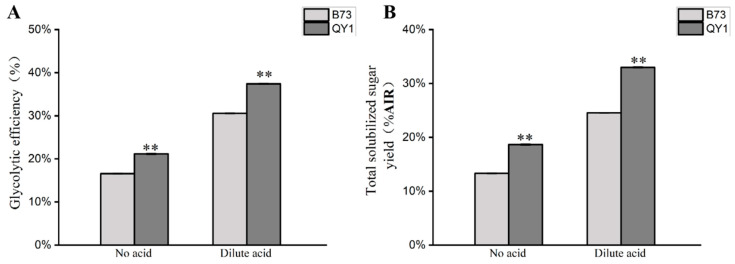
Saccharification efficiency analysis in B73 and QY1 stems. Enzymatic hydrolysis efficiency (**A**) and total soluble sugar production (**B**) of B73 and QY1. ** (*p* < 0.01).

**Figure 5 ijms-25-08195-f005:**
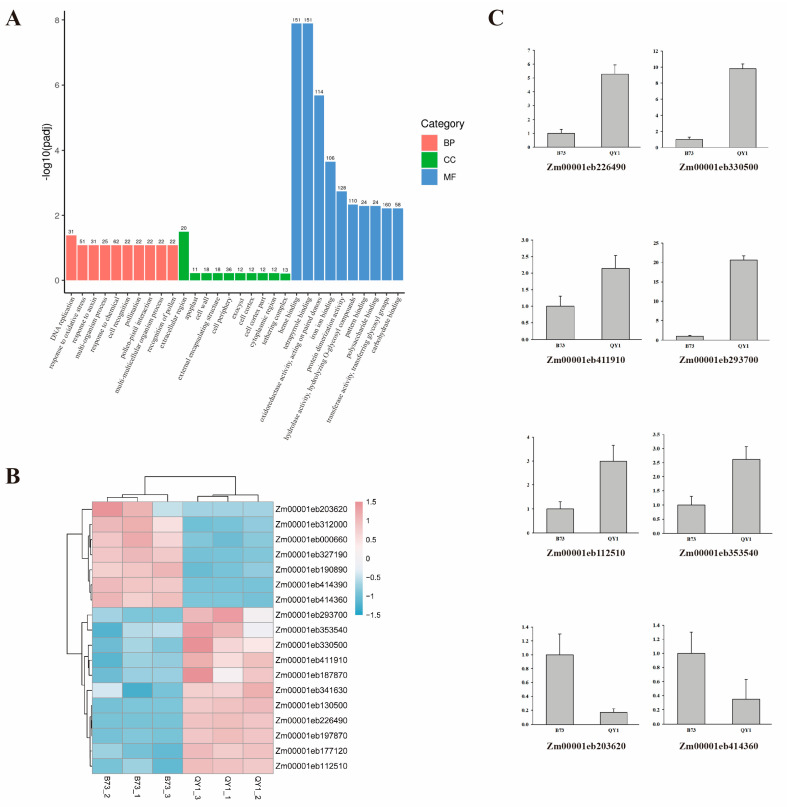
Analysis of DEGs in B73 and QY1 stems. (**A**) GO enrichment analysis of QY1 vs. B73 differential genes. (**B**) Heatmap of cell wall differentially expressed gene analysis. (**C**) qPCR verification of transcriptome differential gene expression.

**Figure 6 ijms-25-08195-f006:**
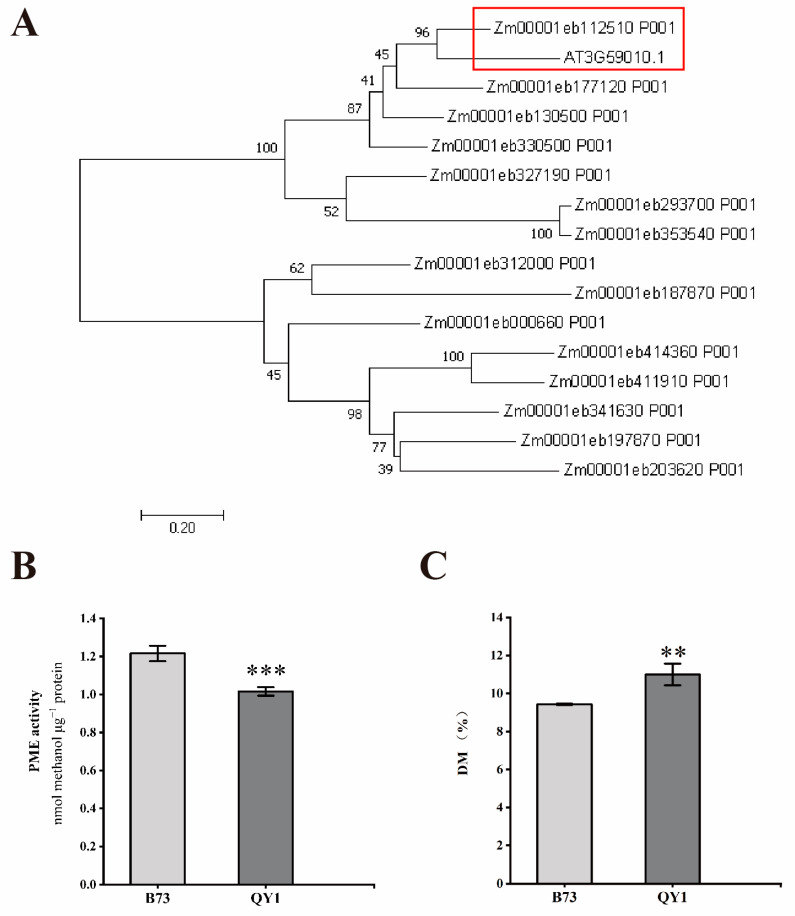
Analysis of PME in B73 and QY1 stems. (**A**) Homology tree of AtPME35 and differentially expressed PME family genes. Analysis of PME activity (**B**) and degree of pectin methyl esterification (**C**) in B73 and QY1 stems. ** (*p* < 0.01), *** (*p* < 0.001).

**Table 1 ijms-25-08195-t001:** QY1 vs. B73 cell wall DEGs.

Gene Number	Comment	log_2_ Fold Change
*Zm00001eb293700*	pectin methylesterase37	4.381
*Zm00001eb130500*	pectin methylesterase19	3.458
*Zm00001eb187870*	xyloglucan:xyloglucosyl transferase activity	3.421
*Zm00001eb330500*	pectin methylesterase20	3.294
*Zm00001eb226490*	xyloglucan endo-transglycosylase/hydrolase4	2.558
*Zm00001eb112510*	pectin methylesterase12	1.612
*Zm00001eb353540*	pectin methylesterase38	1.591
*Zm00001eb197870*	xyloglucan endo-transglycosylase/hydrolase6	1.348
*Zm00001eb177120*	pectin methylesterase15	1.278
*Zm00001eb411910*	xyloglucan endo-transglycosylase/hydrolase2	1.238
*Zm00001eb341630*	xyloglucan:xyloglucosyl transferase activity	1.071
*Zm00001eb190890*		−2.232
*Zm00001eb312000*	xyloglucan:xyloglucosyl transferase activity	−2.363
*Zm00001eb414360*	xyloglucan endo-transglycosylase/hydrolase7	−2.501
*Zm00001eb000660*	ring finger and WD40 repeat3	−2.699
*Zm00001eb327190*	pectin methylesterase41	−4.286
*Zm00001eb203620*	xyloglucan:xyloglucosyl transferase activity	−5.282
*Zm00001eb414390*	xyloglucan endo-transglycosylase/hydrolase1	−6.566

**Table 2 ijms-25-08195-t002:** QY1 vs. B73 cellulose DEGs.

Gene Number	Comment	log_2_ Fold Change
*Zm00001eb097410*	CesA3	7.88167584
*Zm00001eb349960*	CesA3	7.74899559
*Zm00001eb248410*	CesA2	4.276134325
*Zm00001eb046440*	CesA10	−1.139029969
*Zm00001eb097100*	CesA6	−1.99327018
*Zm00001eb370930*	CesA14	−2.746385239
*Zm00001eb370940*	CesA8	−3.092633023
*Zm00001eb043490*	cellulose synthase (UDP-forming) activity	−5.400036017

## Data Availability

Data are contained within the article and Appendix A.

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
