# Peer review of "Identification and Analysis of the Mechanism of Stem Mechanical Strength Enhancement for Maize Inbred Lines QY1"

_ijms, 2024, doi:10.3390/ijms25158195_

Round 1

Reviewer 1 Report

Comments and Suggestions for Authors

The paper addresses the well-known problem of maize lodging, however it seems to be resolved in the practice. More problems is whit the wheat or rye. The finding of the Authors suggest that QY1 could be a valuable resource for developing maize varieties with robust stalk mechanical strength or for use in biofuel production.

The work is planned generally good and the results can be applicable. However, the work should be revised in relation to the structure. The Introduction is comprehensive and can be corrected, but generally this part is good.

Material and Methods should be more detailed and the references should be given.

I don't really see a text that verifies the theses put forward in the paper in Introduction. Essentially, there is a lack of conclusiveness from the results. I don't really see a text that verifies the theses in here. There is a general lack of conclusion from the results. it is difficult to give specific passages here because the work is scattered. The work is regards to the breeding processes, but it is nothing more about it. The maize has generally strong straw. Is there any more factors affected the maize straw? Climate, geographical regions and climatic factors? What about GMO?

Line 72-79: references?

Figure 2: staining;

Figure 3: which staining were used? I think is not toluidine staining where it should be blue/green, but this can be performed with phloroglucinol-HCl solution. It is not clearly described in Methods.

The anatomy figures are not described and indicated on the picture. Scale? Invisible.

What about ,,Resorcinol stain” in Results and discussion?

Line 449-501: ?

Comments on the Quality of English Language

Minor spell check required.

Author Response

Comments 1:The paper addresses the well-known problem of maize lodging, however it seems to be resolved in the practice. More problems is whit the wheat or rye. The finding of the Authors suggest that QY1 could be a valuable resource for developing maize varieties with robust stalk mechanical strength or for use in biofuel production.

The work is planned generally good and the results can be applicable. However, the work should be revised in relation to the structure. The Introduction is comprehensive and can be corrected, but generally this part is good.

Material and Methods should be more detailed and the references should be given.

I don't really see a text that verifies the theses put forward in the paper in Introduction. Essentially, there is a lack of conclusiveness from the results. I don't really see a text that verifies the theses in here. There is a general lack of conclusion from the results. it is difficult to give specific passages here because the work is scattered. The work is regards to the breeding processes, but it is nothing more about it. The maize has generally strong straw. Is there any more factors affected the maize straw? Climate, geographical regions and climatic factors? What about GMO?

Response 1: Thank you very much for taking the time to review this manuscript. We revised the manuscript according to your valuable suggestion. We have added conclusions in every part of results. QY1 could be a valuable resource for developing maize varieties, but in this study, we mainly explain why QY1 has strong stalks. We find that QY1 differs from B73 in terms of  cell wall thickness, vascular bundle area, cell wall composition content ,and the degree of pectin methylesterification. These aspects may be related to the strong mechanical strength of the QY1 stem. Climate, geographical regions, and climatic factors may affect maize stalks, but QY1 exhibits a strong stalk phenotype in different geographical regions. GMOs may also affect maize stalks. We have some maize mutants with brittle stalk phenotypes, and overexpression of related genes exhibits strong stalks (unpublished).

Comments 2:Line 72-79: references?

Response 2: Thank you very much. We have rewritten this part and added references.

Comments 3:Figure 2: staining;

Response 3: Thank you very much,  we have rewritten all the figure legends and replaced all images with high-resolution and correct versions.

Comments 4 :Figure 3: which staining were used? I think is not toluidine staining where it should be blue/green, but this can be performed with phloroglucinol-HCl solution. It is not clearly described in Methods.

Response4 : Thank you for your review. We apologize for our mistakes in the figure legends and methods sections. We have rewritten all the figure legends and methods. This figure shows the Phloroglucinol–HCl stain, and we have revised it in the manuscript. Thank you again.

Comments 5:The anatomy figures are not described and indicated on the picture. Scale? Invisible.

Response 5: Thank you very much. We have replaced all images with high-resolution versions and corrected the scale.

Comments 6:What about ,,Resorcinol stain” in Results and discussion?

Response 6: Thank you for your review. We apologize for our mistakes. The Phloroglucinol–HCl stain is correct, and we have revised it in the manuscript.

Comments 7:Line 449-501: ? 

Response 7:Thank you very much, we have rewritten the Methods section.

Reviewer 2 Report

Comments and Suggestions for Authors

This study analyzed the mechanism of mechanical strength improvement of Maize inbred line QY1. Discovering biomass can be an excellent feedstock for bioenergy and materials, which is significant. In this respect, the topic of this study is interesting. However, it needs some modifications before it can be published.

1. In Figures 1 and 3, some of the text is not readable due to low resolution. This needs to be corrected.

2. Figure 3 presents the cellulose and hemicellulose content (A) and lignin content (H). The units are both % AIR (alcohol-insoluble residue), which analyzes the chemical composition of the biomass after extraction. However, the sum of the cellulose, hemicellulose, and lignin contents is 60% ATR or less. (B73 appears to be lower than 50% ATR.) These are remarkably low. Please check the data or experimental method and correct the data to address the mass loss in terms of mass balance.

3. Is the sum of the monosaccharide contents in Figure 3 (B) the same as the cellulose and hemicellulose contents (A)? In general, they should be approximately the same. If not, the authors need to explain why. Also, try to unify the graphs' units, whether changing the units from ug/mg to %ATR as in (A) and (H).

4. In Figure 4 (B), the yield of total soluble sugar is somewhat lower than in previous studies using biomass of similar species. There should be more explanation in the manuscript as to why this yield was obtained and how it compares to previous studies.

Author Response

Comments:This study analyzed the mechanism of mechanical strength improvement of Maize inbred line QY1. Discovering biomass can be an excellent feedstock for bioenergy and materials, which is significant. In this respect, the topic of this study is interesting. However, it needs some modifications before it can be published.

Response: Thank you very much for taking the time to review this manuscript. We have revised the manuscript according to your valuable suggestions.

Comments 1:In Figures 1 and 3, some of the text is not readable due to low resolution. This needs to be corrected.

Response 1: Thank you for your valuable suggestions. We have replaced all images with high-resolution and correct versions.

Comments 2: Figure 3 presents the cellulose and hemicellulose content (A) and lignin content (H). The units are both % AIR (alcohol-insoluble residue), which analyzes the chemical composition of the biomass after extraction. However, the sum of the cellulose, hemicellulose, and lignin contents is 60% ATR or less. (B73 appears to be lower than 50% ATR.) These are remarkably low. Please check the data or experimental method and correct the data to address the mass loss in terms of mass balance.

Response 2: Thank you for your review. We apologize for our mistakes in the figure legends, and we have rewritten them. Fig. 3A shows the cellulose content, Fig. 3B measures the main monosaccharides that make up hemicellulose and pectin, and Fig. 3H shows the lignin content. Therefore, the sum of Fig. 3A, B, and H represents the cell wall composition. Due to detection errors in each part and limitations of the methods, the sum of Fig. 3A, B, and H is 60-70% of AIR.

Comments 3: Is the sum of the monosaccharide contents in Figure 3 (B) the same as the cellulose and hemicellulose contents (A)? In general, they should be approximately the same. If not, the authors need to explain why. Also, try to unify the graphs' units, whether changing the units from ug/mg to %ATR as in (A) and (H).

Response 3: Thank you for your valuable suggestions. We apologize for our mistakes in the figure legends. Figure 3A and 3B represent two different component contents, and we have rewritten this part of the figure legends. We have also changed the units from µg/mg to %AIR.

Comments 4:In Figure 4 (B), the yield of total soluble sugar is somewhat lower than in previous studies using biomass of similar species. There should be more explanation in the manuscript as to why this yield was obtained and how it compares to previous studies.

Response 4: Thank you for your valuable suggestions. Since we used the basal stem of maize grown in fields, the lignin content is higher, resulting in a somewhat lower yield of total soluble sugar. We have explained this in the manuscript.

Round 2

Reviewer 1 Report

Comments and Suggestions for Authors

The work was corrected, but some of the Figures are still unreadable: the font is to little.

Comments on the Quality of English Language

The work should be read by indepentent person - native-speaker, who can find language mistakes.

Author Response

Comments 1:The work was corrected, but some of the Figures are still unreadable: the font is to little.

Response 1:Thank you for your valuable suggestions.We have replaced the images with higher-resolution versions and reformatted the layout of the images.

Comments 2:The work should be read by indepentent person - native-speaker, who can find language mistakes.

Response 2: Thank you for your valuable suggestions. Our manuscript has been revised by a native English speaker.